# How Single-Molecule Localization Microscopy Expanded Our Mechanistic Understanding of RNA Polymerase II Transcription

**DOI:** 10.3390/ijms22136694

**Published:** 2021-06-22

**Authors:** Peter Hoboth, Ondřej Šebesta, Pavel Hozák

**Affiliations:** 1Department of Biology of the Cell Nucleus, Institute of Molecular Genetics of the Czech Academy of Sciences, Vídeňská 1083, 142 20 Prague, Czech Republic; hoboth@img.cas.cz; 2Faculty of Science, Charles University, Albertov 6, 128 00 Prague, Czech Republic; sebesta@natur.cuni.cz; 3Microscopy Centre, Institute of Molecular Genetics of the Czech Academy of Sciences, Vídeňská 1083, 142 20 Prague, Czech Republic

**Keywords:** cell nucleus, gene expression, transcription foci, transcription factors, super-resolution microscopy, structured illumination, stimulated emission depletion, stochastic optical reconstruction, photoactivation

## Abstract

Classical models of gene expression were built using genetics and biochemistry. Although these approaches are powerful, they have very limited consideration of the spatial and temporal organization of gene expression. Although the spatial organization and dynamics of RNA polymerase II (RNAPII) transcription machinery have fundamental functional consequences for gene expression, its detailed studies have been abrogated by the limits of classical light microscopy for a long time. The advent of super-resolution microscopy (SRM) techniques allowed for the visualization of the RNAPII transcription machinery with nanometer resolution and millisecond precision. In this review, we summarize the recent methodological advances in SRM, focus on its application for studies of the nanoscale organization in space and time of RNAPII transcription, and discuss its consequences for the mechanistic understanding of gene expression.

## 1. Introduction

The genome is a complex and very dense viscoelastic polymer matrix, and therefore, it is difficult to study individual components of the gene expression machinery with conventional light microscopy. The optical resolution of the conventional light microscopy is limited by the diffraction of light and allows to distinguish the objects only if they are ~200 nm apart. This ~200 nm resolution limit of the classical light microscope is given by the nature of light and results from the fundamental laws of physics. Ernst Abbe postulated in 1873 that the limit of discrimination will never pass significantly beyond half the wavelength of blue light [1,2]. According to Abbe’s law of the diffraction limit, the best achievable spatial resolution in the object plane ($d_{x,y}$) and along the optical axis ($d_{z}$) is $d_{x,y}=$ λ/2n sinα (1) and $d_{z}=$ 2λ/n sinα (2), respectively. In (1) and (2), λ represents the wavelength in the vacuum of the light used for imaging and n sinα represents the numerical aperture (NA) of the microscope objective lens, where n is the refractive index of the sample and α is half of the objective lens aperture angle. Hence, reducing the wavelength of the light used for imaging and/or increasing the NA of the objective lens improves the spatial resolution of a conventional light microscope. Lord Rayleigh elaborated on the diffraction limit, and in 1896, postulated that the smallest resolvable distance ($d_{min}$) between two points is proportional to the wavelength in the vacuum of the light used for imaging and inversely proportional to the NA with a factor of 0.61 [3], which is expressed by the formula $d_{min}=$ 0.61 λ/NA (3). According to (3), it is possible to improve the $d_{min}$ by shortening the λ, and therefore, use of electrons instead of light allowed for the visualization of the biological specimens at the ultrastructural level. Many important discoveries of the ultrastructural organization of the eukaryotic cell nucleus, such as the precise localization of transcribed DNA [4] or the ultrastructure of transcription sites [5] were possible thanks to electron microscopy. However, a more detailed summary of the functional organization of the cell nucleus as revealed by EM is beyond the scope of this review and has been reviewed elsewhere [6,7]. Very specific and invasive sample preparation is however the limitation of electron microscopy and renders it impossible for live-cell application.

Super-resolution microscopy (SRM), contrary to electron microscopy, allows for live-cell imaging at the previously unprecedented spatial and temporal level. Novel and progressive SRM approaches helped to uncover many previously undetected dynamic mechanisms that regulate mammalian RNA polymerase II (RNAPII) transcription. Nevertheless, RNAPII is not a solo player but its function is related to the overall gene expression processes and also intimately linked with the steps preceding as well as succeeding the transcription. In this review, we will introduce SRM techniques in general, we will put emphasis on single-molecule localization microscopy (SMLM), and present the variety of fluorophores suitable for SMLM. Then, we will first focus on SMLM applications to studying the dynamics of transcription factors, and finally on the RNAPII itself.

## 2. Brief Introduction to SRM

Although Abbe postulated that the limit of discrimination is given by half the wavelength of blue light, he also acknowledged that this limit could be surpassed by applications based on principles that are outside of the law that he postulated [1,2]. Fluorescent far-field optical microscopy detects light emitted by fluorophores in a specimen using detectors located at a certain distance from the sample. This approach is currently the basis of most biological applications of SRM [8,9,10]. SRM achieves sub-diffraction limited resolution by time-controlled emission of fluorescence from only a subset of fluorophores from the total fluorophore population in the specimen. Two main concepts of SRM are (i) reversible saturable optical fluorescence transitions (RESOLFT) that allow deterministic temporal control of fluorophore emission, and (ii) single-molecule localization microscopy (SMLM) based on the stochastic temporal control of fluorophore emission [2,11,12].

### 2.1. Deterministic SRM Approaches: SIM and STED

The RESOLFT family [8,9,13,14] includes saturated structured illumination microscopy (S-SIM) [15], which is an upgrade of the previously structured illumination microscopy (SIM) [16]. In the classical SIM, Gustafsson implemented high-frequency line-patterned illumination and improved axial and lateral resolution by a factor of two [16,17]. The subsequent 3D SIM improved spatial resolution by an additional factor of two [18,19,20]. However, classical SIM did not demonstrate the potential to increase the resolution without limit, and therefore the classification of SIM as a typical SRM varies [10,11,12]. Nevertheless, S-SIM, which is a nonlinear modification of classical SIM, is in theory, capable of unlimited resolution [15] and is also applicable for live-cell imaging [21]. A typical representative of the RESOLFT family is the stimulated emission depletion (STED) microscopy and its variations [22]. The resolution of a confocal microscope, on which STED is typically based, is increased by attenuating the fluorescence at the periphery of the excitation spot. The fluorescently labeled sample is illuminated by an excitation laser simultaneously with the depletion laser that has a wavelength within the emission spectrum of the imaged fluorophore. The depletion laser beam has a ring or donut shape and is superimposed over the excitation laser beam. The depletion laser at a certain intensity, called saturation intensity, then switches off the fluorescence over a controlled area at the periphery of the excitation spot by a process called stimulated emission [23] and effectively reduces the diameter of a recorded fluorescence spot. Figure 1 shows a visual comparison of histone H2B tagged with SNAP-tag and labeled in living U-2 OS cells with the SNAP substrate JF549, followed by the indirect immunofluorescence labeling of the RNAPII in the same cells upon chemical fixation and imaged by SIM, STED, and dSTORM. SMLM technique dSTORM is introduced in the next section followed by the section on the fluorescent labeling strategies for SMLM.

### 2.2. Stochastic SRM Approaches: Single-Molecule Localization Microscopy

SMLM, in contrast to the above-discussed deterministic SRM techniques, allows for the visualization of individual fluorophores at the level of individual molecules. This revolutionary approach is based on the temporal separation of individual fluorescent molecules based on a relatively simple principle that allows for the detection at subsequent time intervals of a sparse subset of well-separated fluorophores and reconstruction of a final super-resolved image from the localization of the centers of the fluorophores (Figure 2). SMLM techniques originally included photoactivated or fluorescence photoactivation localization microscopy (PALM) [24] or FPALM [25] and stochastic optical reconstruction microscopy (STORM) [26] or direct STORM (dSTORM) [27,28,29] and their modifications. These methods limit the number of fluorophores that emit photons at the same time and assemble the final super-resolved image from sequentially emitting fluorophores. Reduced probability of simultaneous detection of overlapping signals in one acquisition frame allows localizing the center of individual emitters with the precision in the range of lower tens of nanometers, depending on the number of emitted photons. PALM utilizes the photo-activatable fluorescent proteins (PA-FPs) that can be reversibly switched between the non-fluorescent and fluorescent states or the photo-convertible FPs (PC-FPs) that can be switched between two fluorescent states with different excitation and emission pattern [30,31]. Individual molecules of the PA- and PC-FPs are stochastically photo-activated and photo-converted, respectively, while neighboring molecules remain non-fluorescent or undetected and therefore individual FP molecules can be sequentially localized at different time intervals. Fluorescence emitted by the individual and spatio-temporaly well separated PA or PC form of an FP is detected by sensitive cameras and fitting of the Gaussian profile to the signal enables to measure the position of a molecule with a precision far better than in the diffraction-limited conventional fluorescence microscopy. The principle of reversible stochastic photo-switching of organic fluorochromes between the fluorescent and long-lived dark states was first utilized in STORM using photo-switchable dye pairs [26]. This principle allowed for the detection of a sparse subset of fluorochromes in their fluorescent “on” state that is well separated at a given time-point. Thus, the precise localization, e.g., the center of the mass, of individual particles can be determined with high precision and this process is iteratively repeated (Figure 2A). The final super-resolved image is created by the sum of individual single-molecule localization (Figure 2B–D) and therefore resembles of artistic painting technique pointillism (Figure 2E–G) [32]. Soon after its introduction, STORM was implemented for 3D [33,34] and live-cell imaging [35]. In contrast to PALM and STORM that use PA-FPs and a combination of activator and reporter pairs of organic fluorochromes, respectively, dSTORM uses conventional fluorescent probes such as labeled antibodies or self-labeling tags [27,29].

SMLM along the optical plane (*z*-axis) is more challenging than in the imaging (*x*,*y*) plane for several reasons [11,36]. First, out-of-focus fluorescence from above and below the focal plane precludes the precise localization of fluorophores. Second, bleaching of the fluorophores outside of the focal plane reduces localization density. Third, the number of molecules required to adequately sample a given spatial frequency scales exponentially with the spatial dimension and thus making 3D imaging more challenging than 2D imaging. Solutions for 3D SMLM with improved localization accuracy along the *z*-axis utilize astigmatism [33,34], double-plane detection [37], or engineered PSF with the double-helix shape [38].

Wide-field (WF) illumination is commonly used in SMLM but suffers from significant drawbacks [39], such as is the first and second limitations of 3D SMLM mentioned just above. Elimination of those drawbacks significantly improved the single-molecule localization accuracy. Total internal reflection fluorescence (TIRF) microscopy exploits the exponential decay of the evanescent wave created by the complete reflection under an angle larger than the critical angle of a laser beam from the glass surface at the interface between the glass surface and the sample [40]. Thus, the evanescent wave selectively illuminates a region above the glass surface that is up to ~200 nm into the specimen. A highly inclined and laminated optical sheet (HILO) allows for selective illumination of thin regions deeper than the TIRF angle [41]. In contrast to TIRF, HILO uses a laser beam with an incident angle slightly below the critical angle, which creates a slightly inclined light sheet through the specimen but just above the glass surface. HILO illumination increases specific intensity, greatly reduces background, and thereby allows for the precise visualization and quantification of single-molecule dynamics in living cells. More technically demanding modalities for selective plane illumination, such as Gaussian light sheet (LS), require additional illumination objective, which creates a thin sheet of illumination perpendicular to the detection objective [42]. LS illumination permits imaging of thick samples but has limited use for SMLM. A recent combination of LS with the above-discussed RESOLFT neutralized the resolution-limiting role of diffraction in LS. LS-RESOLFT is therefore conceptually diffraction-unlimited and has the potential to develop toward molecular-scale resolution [43]. Bessel beam selective plane (BBSP) illumination overcomes the limitations of Gaussian light sheets and allows for fast imaging across large volumes with increased single-molecule localization precision due to the reduced background fluorescence [44,45]. Lattice light-sheet (LLS) uses ultrathin light sheets derived from 2D optical lattices that are scanned through the specimen to image across large volumes with high spatial and temporal resolution and minimal photo-damage [46]. In the reflected light sheet (RLS) microscopy, a mirror reflects the elliptical laser beam and thereby creates a thin sheet of light parallel with the imaging plane, which allows for the selectively optical sectioning (<0.5 um) throughout the cell nucleus. Selective illumination of <0.5 µm plane results in the photoactivation of only a small subset of photoactivatable fluorescent proteins, and in turn, dramatically increases the signal-to-noise ratio critical for the direct monitoring of the single-molecule kinetics [47,48].

SMLM and STED have recently met at their intersection called minimal photon emission fluxes (MINFLUX). This novel concept combines stochastic switching of the fluorophores on and off as in PALM or STORM but the emitter is localized by a doughnut-shaped excitation beam that is used in STED [49]. As a result, MINFLUX and its derived fluorescence nanoscopy techniques require up to orders of magnitude fewer photon emissions to deliver single-digit nanometer spatial resolution and 100-times improved temporal resolution as compared to its parent imaging approaches [50]. MINFLUX and its modifications are applicable in living cells for 3D multicolor imaging and microsecond tracking [51,52] and await further explorations.

In this review, we will further discuss the developments of the fluorescent labeling strategies for SMLM and then their use together with the above-reviewed SRM techniques to reveal the novel paradigms in the regulation and dynamics of RNAPII transcription.

## 3. Progressive Development of Fluorophores for SMLM

The rapid development of live-cell SMLM applications is linked to the progressive development of fluorophores with specific and unprecedented photophysical properties. Fluorophores used in SMLM can either be genetically encoded FPs, “self-labeling” tags such as HaloTag or SNAP that covalently associate with organic fluorochromes or organic fluorochromes such as Alexa Fluores attached directly to a primary antibody or attached to a secondary antibody directed against the primary antibody as in classical immunofluorescence protocols (Figure 1 and Figure 2). A plethora of conventional fluorophores, such as Alexa Fluor 647 that is the most commonly used in dSTORM [35,53], are suitable for SMLM [39]. The advantage of rhodamine-based fluorophores is that they exist in a dynamic equilibrium between a fluorescent zwitterion and a non-fluorescent but cell-permeable spirocyclic form. This feature of rhodamine dyes and their derivatives can be differently utilized in various SRM modalities that have different requirements of the dynamic equilibrium between the zwitterion and spirocyclic rhodamine forms [54,55].

As mentioned above, individual fluorescent proteins are randomly photo-activated or photo-converted at different time points in PALM, while their neighbors remain dark or undetected [24,25,30]. The monomeric members of the Eos family of the PC-FPs [54] provide high brightness with good contrast and are suitable for live-cell SRM [24,30]. Photo-convertible fluorescent protein mEos2 [56] is an improvement of its parent protein mEosFP [57] that was not suitable for use in mammalian cells due to its maturation at lower temperatures. Utilization of mEos2 facilitated, for instance, the single-molecule kinetics study of transcription factors (TFs) glucocorticoid receptor (GR) and estrogen receptor-α (ER) [47]. Photo-convertible fluorescent protein Dendra2 [58] is initially green-emitting and upon 405 nm illumination, converts into a red-emitting form [59,60]. Illumination with the very low (~1 W/cm^2^) intensity of 405 nm light allows for the photo-conversion of only a small subset of Dendra2 molecules in the sample. Favorable photo-physics [61] and low aggregation propensity [62] of Dendra2 made it optimal to study, for instance, the dynamics of RPB1, the large catalytic subunit of RNAPII [63,64,65,66], transcription coactivator Mediator [66] or TFs c-Myc and P-TEFb [67]. However, the photo-converted Dendra2 molecules remain in the red-emitting state for several tens or even hundreds of milliseconds, and therefore, a single molecule may appear in multiple acquisition frames. Moreover, in addition to irreversible photo-bleaching, Dendra2 molecules undergo intermittent photo-physical blinking transitions [61,68] that obscure direct correlation between counts of detections and exact numbers of molecules [64].

Genetically encoded self-labeling protein tags such as HaloTag and SNAP [69,70,71] have enzymatic activity through which cell-permeable fluorescent organic dyes that can also be photoactivable covalently attach. HaloTag and SNAP tagged proteins exhibited stable binding events in the nucleus [67] but these unspecific binding events were separated from the specific binding events of the proteins of interest [72]. A comparative SRM study of tagged TF favored HaloTag in terms of unspecific binding, the photostability of the conjugated fluorophore, or localization precision [73]. The discovery of a fine-tuning method for the development of new and highly photostable fluorophores called Janelia Fluors (JFs) [74] for live-cell SRM revolutionized the field and opened almost unlimited opportunities to explore living systems at high spatial and temporal resolution. JFs are available as reactive organic molecules for direct labeling or as HaloTag and SNAP substrates. An important advantage of many of the JFs is their fluorogenicity, e.g., a significant increase in fluorescence upon their binding to HaloTag or SNAP, which alleviates the necessity of extensively washing out unbound fluorophores and reduces unspecific background [75]. Some of the most frequently used JFs so far (Table 1) were JF549, which is a prototypical JF and a direct analog of tetramethyl rhodamine (TMR) or its silicone-containing far-red counterpart JF646 [74,76]. Photoactivatable versions of JF549 and JF646 allow to switch on only a subset of fluorophores, which is advantageous for single-particle tracking (SPT) in living cells [77].

The development of PA fluorochromes extended the possibilities previously offered by PA- or PC-FPs. However, the dependence on the photoactivation for SMLM and SPT can be genetically circumvented by the precise copy number control of the fluorescently labeled fusion proteins [78]. This approach is particularly useful for the imaging of individual molecules in the densely packed nucleus. Progressive development of finely-tuned fluorophores [55] including fluorophores with very specific features, such as photosensitizers that generate reactive oxygen species [79] for delicate applications and allowing for multiplexing at a high spatial and temporal resolution [54] continues to push the boundaries of SMLM applications. Moreover, the fluorescent signal of individual molecules of interest can be amplified by multivalent tags such as Sun-tag [80] or repetitive HaloTag [71,78].

Taken together, the rapidly growing selection of highly specific fluorescent markers with superb properties binding to or being expressed as a part of individual molecules makes SMLM the method of choice to study the functional architecture of the cell nucleus and dynamics of transcription within its spatial context. Some of the key applications of fluorophores for SMLM studies of the RNAPII transcription and associated molecular processes are summarized in Table 1. We will further discuss in the following sections how SMLM expanded our understanding of RNAPII transcription using several studies of TF and RNAPII dynamics as examples.

## 4. Revised Model of Transcription Pre-Initiation Based on the TF Single-Molecule Kinetics

Diffusion of TFs allows them to search for their targets to initiate the transcription of specific genes and thereby regulate patterned gene expression [81,82,83]. Earlier studies of the TF diffusion, mostly performed using fluorescence recovery after photobleaching and similar methods [84], enabled quantification of the dynamics of TF subpopulations but underestimated the impact of nuclear architecture on TF dynamics and did not reveal the DNA binding kinetics of individual TF molecules [85,86,87]. Implementation of SMLM for the investigation of the TF dynamics highlighted that the interaction of regulatory proteins with target sites in chromatin is an even more dynamic process than previously thought, which led to the revision of the model in which the pre-initiation complex assembles by subsequent recruitment of its individual components [88].

Several TFs were recently scrutinized by SMLM, which helped to uncover the common as well as distinct dynamic patterns, explaining the universal mechanisms as well as the roles of distinct TFs in distinct transcriptional programs. In general, the TF kinetics at the single-molecule level was inferred either by fast time-lapse imaging and SPT [89,90] or by blurring off the fast-diffusing molecules and localization of residing TFs at long acquisition times. Stochastic labeling and long-time SPT of JF549-labelled Sox2 tagged with 3-repeats of HaloTag and imaged in living mESCs by time-lapse microscopy using HILO illumination revealed that Sox2 dynamically hops and interacts locally in the nucleus [78]. This study underscored that functional architecture in the nucleus kinetically facilitates local exploration of Sox2, confined Sox2 target search, and thereby facilitated Sox2 activity in the regulation of the transcription of its target genes.

SPT in 2D of TMR-HaloTag-Sox2 visualized by epi- and BBSP illumination [44,45] in living mESCs revealed two populations of Sox2 molecules with respect to their dwelling time on DNA [72]. Short-lived population (<1 s) represented non-specific binding of Sox2 to DNA, while long-lived population (>10 s) represented Sox2 molecules bound to their specific DNA target sequences. Multifocus microscopy (MFM) that allows high-resolution instantaneous 3D imaging [91] combined with 3D SPT and kinetic modeling uncovered that Sox2 and Oct4 search for its targets by a trial-and-error sampling mechanism. This mechanism consisted of 3D diffusion events lasting several seconds and interspersed by a brief (<1 s) non-specific collision before dwelling at specific target DNA for >10 s. The in vitro assay in which TMR-HaloTag-Sox2 slide along the DNA in 1D complemented the data from living cells and showed that sliding of Sox2 in 1D along open DNA facilitated its 3D diffusion-dominated search for the target DNA followed by assisted binding of Oct4. Oct4 then helps to stabilize the interaction of Sox2 with DNA but has little ability to assist Sox2 in its target search. Instead, the Sox2-assisted binding of Oct4 and subsequent binding of Oct4 to Sox2-DNA complex stabilizes the association of the Sox2-Oct4 complex with its target DNA [72]. This detailed analysis facilitated by the recent developments in SMLM is in agreement with the previous notion that the interaction between Sox2 and Oct4 is DNA-dependent, it further revealed the hierarchically ordered dynamic search mechanism of the enhancer-binding pluripotency regulators Sox2 and Oct4, and mechanistically explains the enhanceosome assembly.

The fact that only a small (3%) subpopulation of Sox2 molecules is bound to DNA at any given time [72] makes it impossible to infer the spatial distribution of Sox2 enhancer sites simply from fluorescence fluctuations captured by WF imaging or conventional SRM of fixed cells. Those limitations were overcome when LLS imaging and SPT of JF549-HaloTag-Sox2 were implemented to study the precise 3D distribution of Sox2 enhancer sites in the whole nucleus of a living mESC [92]. This study revealed that Sox2 bound enhancers are not uniformly distributed throughout the nucleus but display dramatic clustering behavior and form locally enriched distinct higher density clusters (EnCs) characterized by high-density clustering of Sox2. Probing of the spatial relationship between Sox2 EnCs and heterochromatin (HC) marker HP1-GFP that forms non-diffraction limited structures in the nucleus revealed that the location of EnCs and HC is only very weakly correlated. Specifically, dual-color WF imaging, mapping of the stable Sox2 EnCs by SPT followed by 2D kernel density estimator, and creation of an EnCs intensity map in the nucleus that was superimposed with the HP1-GFP. This allowed for the quantitative pixel-to-pixel correlation between EnCs and HC, which was confirmed by pair cross-correlation analysis (PCC). PCC, which examines the degree of co-clustering and co-localization between two types of molecules [93], showed no apparent spatial correlation between EnC and HC intensity maps [92]. Additionally, LLS imaging and SPT analysis confirmed and further strengthened the notion of spatial separation between Sox2 EnCs and HC in 3D and showed lower levels of Sox2 in HC compared to surrounding sub-nuclear regions. Finally, dual-color labeling of one target, Halo-Sox2, in the same living mESC with two dyes, JF549 and JF646, and WF imaging of each dye under different conditions, enabled simultaneous visualization of Sox2 EnCs and the dynamics of diffusing and binding Sox2 molecules. Mapping of stably bound Sox2 in EnCs was performed by low excitation and long acquisition times of JF-646-HaloTag-Sox2, while tracking of the fast diffusing and binding dynamics of JF549-HaloTag-Sox2 was performed using high excitation power and short acquisition times. This analysis revealed that most of the Sox2 molecules in EnCs were in a bound state [92].

The single-molecule kinetics of steroid receptors, such as glucocorticoid receptor (GR) tagged with PC-FP mEos2 in living MCF-7 cell line was facilitated by RLS microscopy [47]. The diffusional characteristics of mEos2-GR and other TF, estrogen receptor-α were inferred on the principle of detection by localization [94]. Detection by localization does not rely on the detection of long continuous traces and suggested almost a magnitude lower DNA residence time (RT) of GR than those earlier obtained by FRAP [95]. This finding documents previously unappreciated high dynamics of GR and underscores the advantage of high spatial and temporal resolution of SMLM for studying TF kinetics.

Kinetic studies of yet other TFs, protooncogene c-Myc, and elongation factor P-TEFb revealed that despite their similar diffusion rates, each TF explores the space differently, highlighting their distinct roles in the regulation of transcription [67]. SPT and detailed quantitative analysis of the diffusion kinetics of Dendra2-tagged c-Myc and P-TEFb, visualized by HILO illumination and precisely controlled photoconversion of Dendra2 allowed for the selective visualization of individual Dendra2-tagged c-Myc and P-TEFb and statistical evaluation of their dynamics. Diffusion rates of TFs exceed 10 µm^2^/s and therefore the localization of individual TF molecules deviates from the classical Gaussian PSF. Izeddin et al. devised an algorithm that allowed continuous tracking of TFs diffusing at rates exceeding 10 µm^2^/s. This analysis revealed that although the dynamics of both TFs were similarly restrained by their interaction with nuclear structures, c-Myc and P-TEFb adopt different strategies for the exploration of nuclear space. While Dendra2-tagged c-Myc globally explores the nucleus and diffuses similarly to free Dendra2, individual Dendra2-tagged P-TEFb molecules explored the nuclear volume by sampling space of reduced dimensionality and displaying the characteristic of exploration constrained by fractal structures [67].

The discovery of the hierarchical and dynamic assembly mechanism of the general TFs suggests that this previously undetected mechanism of RNAPII pre-initiation is universal [96]. In vitro single-molecule imaging studies allowed studying of the kinetics of individual purified transcription factors and revealed their stepwise assembly into the pre-initiation complex in a defined environment [97,98]. The in vitro studies were supported by fast-tracking of multiple TFs simultaneously in living cells with high spatiotemporal resolution using a high-sensitivity multicamera system with stroboscopic illumination [99]. In living cells, general TFs bound promoter dynamically and transiently and RNAPII pre-initiation complex assembled hierarchically [96], similar to the enhanceosome [72].

Taken together, the single-molecule imaging uncovered previously unknown Sox2 EnCs positioning throughout the 3D space inside the nucleus and the likelihood of Sox2 finding and binding to EnCs. These detailed quantitative analyses of the Sox2 dynamics and spatial distribution of Sox2 EnCs provided valuable insight into the spatial organization and dynamics of the enhancer-binding pluripotency regulators in embryonic stem cells, which greatly extended our mechanistic understanding of the spatial and temporal regulation of gene expression in pluripotent cells and mammalian development at a molecular level. Moreover, the exemplar studies reviewed above underscored the advantages of single-molecule imaging and revealed previously undetected hierarchical transient-to-stable transition of TF-binding dynamics that regulate the assembly of RNAPII pre-initiation complex. In the next section, we discuss the recent progress in the understanding of RNAPII transcription dynamics with the examples of several detailed single-molecule imaging studies.

## 5. Dynamics of RNAPII by SMLM

Transcription by RNAPII is essential for gene expression and therefore fundamental for cellular activities. RPB1 enzyme, the large catalytic subunit of RNAPII central to the transcription process, forms spatially compartmentalized foci [100,101,102]. These foci, earlier visualized by a combination of light and cryo-electron microscopy [4] are visible in the chemically fixed cells immunolabeled by anti-RPB1 C-terminal domain (CTD) primary antibody followed by secondary antibody labeled by Alexa Fluor 647 and imaged by dSTORM (Figure 2). It is worth mentioning that the antibody used in Figure 2 recognizes all RPB1 CTD forms, however, a plethora of antibodies characteristic for specific stages of transcription exists. Human RPB1 CTD contains 52 heptad repeats that can be reversibly phosphorylated and the specific phosphorylation pattern reflects subsequent steps of transcription [103,104,105], Importantly, distinct steps of transcription could be therapeutically targeted in cancer and other complex diseases [106]. Immunolabeling with antibodies against distinct RPB1 CTD phosphorylation results in distinct spatio-temporal localization of RPB1 and reflects distinct stages of transcription. Nevertheless, multiple antibodies can bind to one RPB1 CTD due to its repetitive organization. Tagging of individual RPB1 molecules with an FP or self-labeling tag, such as HaloTag or SNAP-tag, represents more accurate approaches. Replacing of endogenous RPB1 in living cells with RPB1 fused to PA-FP Dendra2 or tagging of endogenous RPB1 by CRISPR/Cas9 with Dendra2 or HaloTag and allowed to follow the dynamics of individual RPB1 molecules by SMLM in cell lines and embryonic stem cells [63,65,107]. The analysis of real-time dynamics of Dendra2-RPB1 in human U-2 OS cells by time-correlated (tc) PALM showed that RPB1 in living cells exhibits transient dynamic clustering clearly distinct from static clusters. This analysis of the temporal sequence of individual RPB1 molecule detections leading to the formation of a particular RPB1 cluster favors the self-organization model of highly dynamic de novo clustering of RPB1 over the deterministic organization of static RPB1 clusters [63]. The half-life of RPB1 clusters was in the order of seconds [63], which is significantly faster than the several minutes required to complete the transcription of a typical mammalian gene. Further extending the tcPALM analysis to a dual-color mode in living mouse embryonic fibroblasts (MEFs) allowed for the simultaneous visualization of RPB1 and nascent mRNA transcript [64,65]. Simultaneous single-molecule imaging of Dendra2-RPB1 and nascent mRNA containing MS2 hairpin loops recognized by Halo-MCP [64,108,109] labeled with Halo-substrate JF646 by tcPALM and STORM, respectively, revealed that the number of nascent mRNA molecules directly correlates with the lifetime of RPB1 clusters. Transcription activation led to the increased lifetime of RPB1 clusters but not of their frequency, which indicates increased loading of RNAPII molecules onto the gene. The quantitative analysis of the single-molecule dynamics of RPB1 clusters and nascent mRNA supported by theoretical modeling is, however, inconsistent with elongating RNAPII, because the time required for the production of a single mRNA molecule is much longer than the lifetime of RPB1 clusters measured by tcPALM. This notion is nevertheless in agreement with another study that used live-cell SPT of Halo-RPB1 labeled with PA-JF549 and imaged by stroboscopic photoactivation (SPA) [107]. This study showed that the length of CTD, e.g., the number of heptad repeats, substantially influences RPB1 dynamics. Shorter CTDs, such as 26 heptad repeats present in *S. cerevisiae*, resulted in the higher RPB1 cluster dynamics and artificial lengthening of RPB1 CTD led to the lower dynamics of RPB1 clusters and their increased association with chromatin [107]. Intriguingly, RPB1 CTD shortening in U-2 OS cells delayed transcription activation [110], which is in line with the evidence obtained by SMLM that RPB1 CTD interacts in vivo with TF hubs [111]. Importantly, Boehning and co-workers [107] showed that upon its CTD phosphorylation and switching into the elongation phase, RPB1 clusters dissolve, indicating that elongating RNAPII leaves clusters formed at the (pre-) initiation phase [107]. The notion of the individual RNAPII molecules that transcribe genes and produce mRNA in the elongation phase is in agreement with at least two previous studies. In the first study, the authors probed by RLS-SRM the spatial organization of RPB1 tagged with SNAP-tag and labeled by TMR SNAP-substrate [48]. Another study by Bayesian localization microscopy (BLM) [112] that allowed the direct observation of the assembly and disassembly of Dendra2-RPB1 also revealed RNAPII clusters on the actively transcribed genes in the pre-elongation phase [113]. Therefore, the dynamic clustering of RPB1 is linked to the assembly of macromolecular complexes during pre-initiation and initiation steps of transcription. Short-lived clusters observed on the actively transcribed gene are transient agglomerations of ~80 polymerases, which are distinct from the much lower counts of elongating polymerases that likely appeared as low frequency of detections [64].

**Table 1 ijms-22-06694-t001:** Selected exemplar SMLM approaches of tagged TFs and RNAPII in model systems. Abbreviations are explained in the text.

Imaging Approach	Utilized Fluorophore	Target Molecule	Model System	Reference
HILO, SPT	3xHaloTag-JF549	Sox2	mESCs	[78]
BBSP 2D SPT MFM 3D SPT	HaloTag-TMR	Sox2, Oct4	mESCs 3T3 cell line	[72]
PALM, SPT	Dendra2	c-Myc, P-TEFb	U-2 OS cell line	[67]
LLS, SPT, PCC	HaloTag-JF549	Sox2	mESCs	[92]
RLS, RT	mEos2	GR	MCF-7 cell line	[47]
LLS tcPALM	Halo-JF549/646	RPB1 Med	mESCs	[66]
PALM	Dendra2 HaloTag-JF549	RPB1 Sox2	mESCs	[92]
tcPALM	Dendra2	RPB1	U-2 OS cell line MEFs	[63] [64,65]
SPA-SPT	Halo-PA-JF549	RPB1	U-2 OS cell line	[103]
RLS	SNAP-TMR	RPB1	U-2 OS cell line	[48]
BLM	Dendra2	RPB1	U-2 OS cell line	[108]

Spatial relationship and inferred functional correlation of RNAPII distribution and Sox2 enhancers was investigated in living mECSs by 2D SPT and mapping of Sox2 EnCs (as described in the previous section) by low-excitation and long-acquisition detection of stable DNA bound JF646-HaloTag-Sox2 molecules followed by PALM imaging of photoconverted Dendra2-RPB1 [63,92]. Auto-correlation analysis revealed that RPB1 molecules displayed local density fluctuations, e.g., clustering of RPB1 [92], consistent with previous reports [63]. However, RPB1 clusters were more evenly distributed throughout the nucleus and had lower packing densities than highly clustered Sox2 EnCs that also displayed tighter packing [92]. Quantification of the spatial relationship between the distribution patterns of RPB1 and Sox2 by the pixel-to-pixel correlation between the intensity of RPB1 and Sox2 EnCs in individual cells showed that EnC regions are generally correlated with RPB1 occupancy and pair cross-correlation of Sox2 EnCs with RPB1-enriched regions showed a significant degree of co-localization/clustering. However, due to the tighter clustering of Sox2 enhancers, most Sox2 EnC regions contained significant levels of RPB1, whereas only a subset of RPB1 enriched regions overlapped with Sox2 EnCs [92], which is in agreement with the previous notion that Sox2 only targets a subset of genes transcribed in mESCs [114].

The dynamic clustering of RPB1 before productive transcription elongation revealed by tcPALM and SPT analyses has implications for the fine regulation of gene expression and was further confirmed by detailed quantitative analysis of the assembly of macromolecular complexes during pre-initiation and initiation steps of transcription. Transcription coactivator Mediator (Med) assists in the assembly of a transcription pre-initiation complex. Endogenous Med tagged with Dendra2 forms similarly to RPB1 small (~100 nm) transient (lifetime ~10 s) and large (>300 nm) stable (lifetime > 100 s) clusters in living mouse embryonic stem cells (mESC). The differentiation of mESCs into epiblast-like cells (EpiLCs) had no apparent effect on the small transient Med clusters but reduced the size and the number of large stable Med clusters [66]. Dual-color lattice light-sheet imaging of JF646-HaloTag-Med followed by imaging of Dendra2-RPB1 in mESCs revealed that the vast majority (90%) of large stable Med clusters also contained RPB1. The association between Med and RPB1 appeared to be hierarchical with Med clusters serving as a scaffold for the RPB1 recruitment. Single-color LLS-SRM of JF646-HaloTag-RPB1 or JF646-HaloTag-Med dynamics that allowed fast (order of seconds) volumetric acquisitions of the whole nucleus in living mESC and precise 4D tracking revealed high mobility and coalescence of individual RPB1 or Med clusters into larger condensates. Moreover, dual-color imaging of JF549-HaloTag-Med together with a gene loci visualized by mRNA-incorporated MS2 hairpin that binds MCP-HaloTag-JF646 [64,108,109] revealed the mobility and coalescence of individual RPB1 or Med clusters and transient co-localization of Med with the gene loci during transcription pre-initiation [66]. Taken together, LLS-SRM [92] that allows gentle multicolor 4D imaging of individual molecule dynamics in living cells helped to elaborate on the previous measurements of single-molecule dynamics by tcPALM. These individual techniques separately revealed the induced condensation of Med at enhancers followed by recruitment of RNAPII and preceding mRNA production [64], which is consistent with the dynamic recruitment of Sox2 and Oct4 to their enhancers [66,92] that we discussed in the preceding section.

In summary, several reviewed SMLM approaches enabled us to study the RNAPII transcription in unprecedented spatial and temporal details and thereby helped to finely refine our mechanistic understanding of the regulation of gene expression. Integration of the information obtained by the SMLM together with biochemistry helped to build a mechanistic model of the localized hierarchical recruitment of co-activators and TFs during the pre-initiation phase followed by the recruitment and formation of RNAPII clusters during the initiation phase and followed by RNAPII leaving the clusters during elongation phase. This hierarchical and dynamic principle of the transcriptional regulation and the discovery of RNAPII forming transient condensates that dictates the productive transcriptional output has progressively changed our mechanistic understanding of gene expression.

## 6. Conclusions and Perspectives

Since the publication of one of the seminal reviews on SRM in 2010 [11], twelve years have passed during which the expectation of authors that “it will still take time and further engineering until (the) technical developments (of SMLM) find their way into commercial systems” has become a reality. Moreover, as we have reviewed here the examples of some of the SMLM applications to monitor the dynamics of TFs and RNAPII in the living cell, in the past decade we have witnessed rapid progress in the 4D single-molecule kinetic studies in living cells. Nevertheless, many areas of the cell nucleus that hold secrets of the gene expression mechanism that lead to the establishment and propagation of the living matter in space and time remain unexplored. We believe that the relatively new but rapidly growing field of nuclear lipid biology will greatly benefit from the progressive development of quantitative single-molecule imaging approaches. Nuclear lipids and phosphatidylinositol phosphates (PIPs) in particular, form small foci variably distributed in the eukaryotic cell nucleus (Figure 3), localized to the specific sub-nuclear compartments, and linked with the crucial nuclear processes [115,116,117,118,119,120,121,122,123]. So far, we have started to paint with nanometer precision the static pictures of nuclear PIP distributions [124,125]. However, as we learned from the cytoplasmic membrane, intracellular membrane compartments, and nuclear envelope, lipids are highly mobile entities [52]. Although a variety of tools is applicable for the specific visualization of various lipid species in the membranes, lipids in the membranes display 2D distributions and dynamics that differ from the nuclear interior that is organized within membrane-less sub-compartments [126]. Therefore, the development of the tools for visualization of specific nuclear lipids by single-molecule imaging together with the use of novel fluorophores and implementation of the 3D SPT will be crucial for the kinetic studies of the nuclear lipids and their roles in the establishment of the functional nuclear architecture, regulation of transcription and overall gene expression. Tracking of the individual molecules of specific lipid probes in the compact nuclear environment will be guided by the SPT of transcription factors. The combination of SMLM with EM provided valuable contextual information about cellular nanostructures [24,127] and hold potential for the unraveling of the lipid and in particular PIP, the identity of the nuclear gene expression compartments.

## Figures and Tables

**Figure 1 ijms-22-06694-f001:**
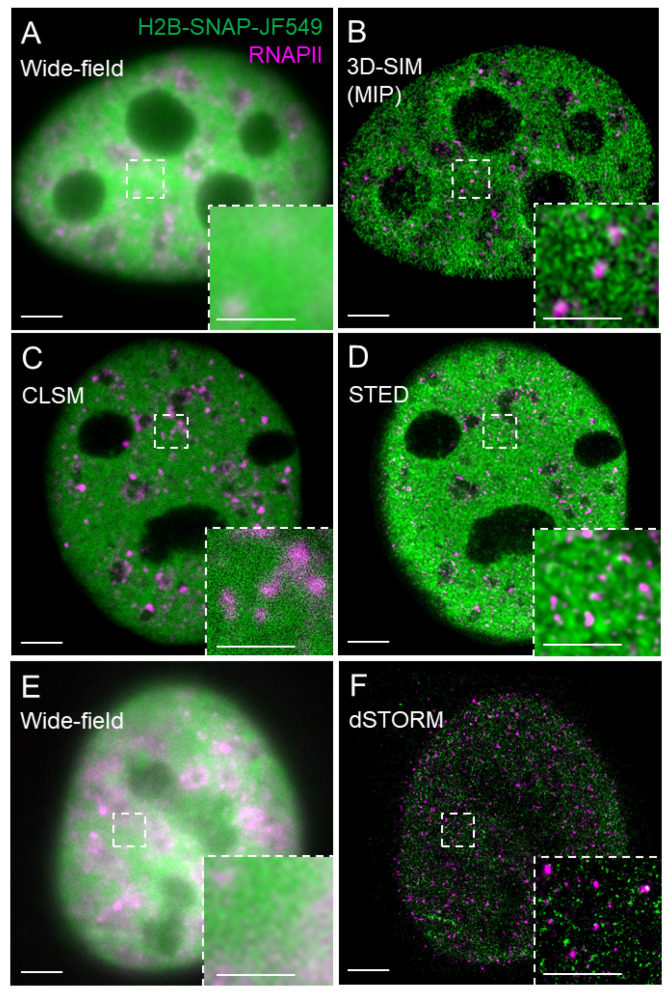
Visual comparison of SIM, STED, and dSTORM. Nuclei of U-2 OS cells expressing histone H2B tagged with SNAP-tag were labeled with the JF549 SNAP substrate (H2B-SNAP-JF549, green), chemically fixed, and indirectly immunolabeled against RNAPII with a secondary antibody conjugated with Alexa Fluor 647 (magenta, Abberior STAR 635P for STED). The same cell was imaged by (**A**) wide-field and (**B**) 3D-SIM (MIP—maximum intensity projection of a 3D-SIM z-stack); (**C**) confocal laser scanning microscopy (CLSM) and (**D**) STED microscopy followed by deconvolution; (**E**) wide-field and (**F**) 2D dSTORM. Construct H2B-Halo-SNAP and Janelia Fluor (JF)549 SNAP tag substrate were kind gifts of Xavier Darzacq from UC Berkeley and Luke Lavis from Janelia Research Campus of the Howard Hughes Medical Institute, respectively. Bar = 5 µm; inset 1 µm.

**Figure 2 ijms-22-06694-f002:**
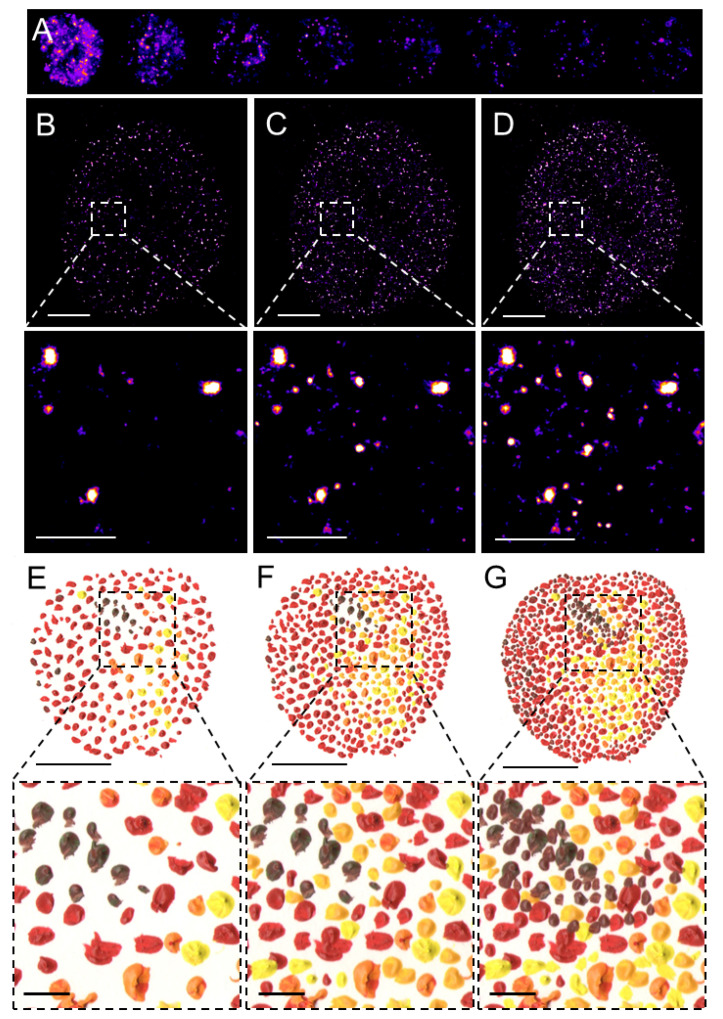
Pointillistic nature of single-molecule localization microscopy. (**A**) Individual frames from the time-lapse acquisition of raw blinking of Alexa Fluor (AF)647-labeled secondary antibody against primary anti-RNAPII antibody (RNAPII-AF647). The signal of individual AF647 molecules accumulates in time (**B**,**C**) to create the final dSTORM image of RNAPII (**C**). The imaging process in (**B**–**D**) resembles an artistic painting style called pointillism in which the accumulation of color spots on a canvas (**E**,**F**) creates the final painting (**G**). Artistic paintings in (**E**–**G**) were kindly provided by Katarína Mrvová. White bar = 5 µm; inset 1 µm. Black bar = 5 cm; inset 1 cm.

**Figure 3 ijms-22-06694-f003:**
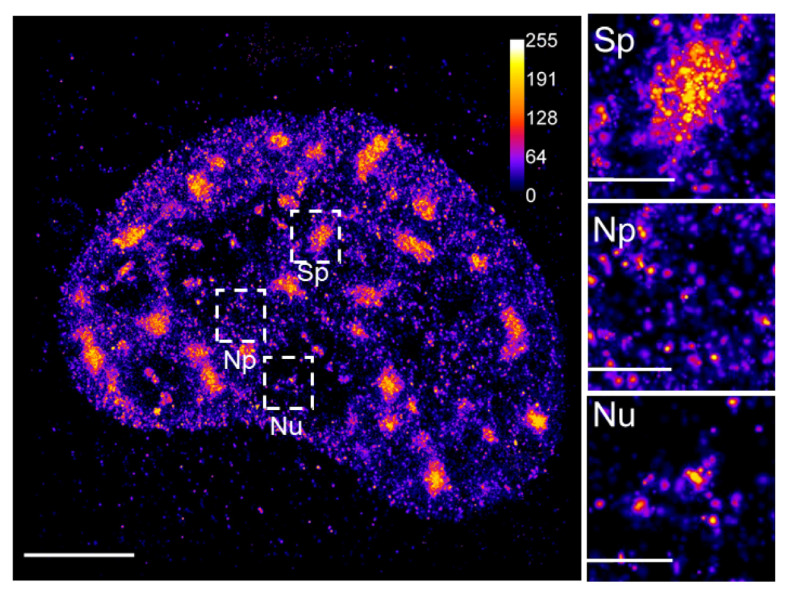
3D dSTORM imaging of nuclear phosphatidylinositol 4,5-bisphosphate. A 2D projection of the 3D dSTORM image of the nuclear phosphatidylinositol 4,5-bisphosphate (PIP2) indirectly immunolabeled by primary anti-PIP2 antibody followed by secondary antibody conjugated with Alexa Fluor 647 and imaged by 3D dSTORM. Selected regions show specific pattern of PIP2 associated with nuclear speckles (Sp), nucleoplasm (Np) and nucleolus (Nu). Calibration bar shows relative fluorescence intensities. Scale bar = 5 µm; inset 1 µm.

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
