# Peer review of "How Single-Molecule Localization Microscopy Expanded Our Mechanistic Understanding of RNA Polymerase II Transcription"

_ijms, 2021, doi:10.3390/ijms22136694_

Round 1
Reviewer 1 Report
In this review Hoboth et al. present an overview of recent advancements of single molecule imaging techniques and their impact on forming a precise spatio-temporal image of RNA polymerase II transcription within nuclei. The review is designed to first provide a technical overview (history, types of SRMs, fluorophores) followed by an overview of the impact of SRM methods of our current view on the steps regulating gene transcription.
The rapid development of SRM methodology certainly makes it a timely review focusing on SRM-based research on pol II-driven transcription. The review is well designed and written and gives a good overview of the field. Thus, it will be relevant for a wide range of colleagues in the field. I can recommend publication in its current form but would ask to address the following very minor concerns:
Line 56: I understand what the authors mean by emphasizing “gene expression”. However, this review is essentially about transcription and I suggest to keep the focus on “transcription” rather than on the broad term gene expression.
Line 70: the sentence is difficult to understand (“many wavelengths away from the sample”), please modify.
Line 402: in the light of the following discussion it may be worthwhile mentioning here that antibodies exist specific to different post-translational modifications of RPB1 with very distinct biological functions. That way, the reader can more easily apprehend what the authors mean when using terms like “foci” in context of pol II assembly, binding or elongation.
Author Response
Dear Reviewer,
We appreciate your positive evaluation and thank you for your suggestions that helped us to improve our manuscript.
In the revised version of our manuscript we replaced the term "gene expression" in line 56 as you suggested and everywhere where applicable for the term "transcription" to keep it specifically in the line with the focus of our review. The rephrased sentence in line 56 now reads: "Novel and progressive SRM approaches helped to uncover many previously undetected dynamic mechanisms that regulate mammalian RNA polymerase II (RNAPII) transcription." Then follows the original text: "Nevertheless, RNAPII is not a solo player but its function is related to the overall gene expression processes and also intimately linked with the steps preceding as well as succeeding the transcription.", followed by the slightly rephrased: "In this review we will introduce SRM techniques in general..."
We have rephrased the sentence in line 70 to make it more understandable. The sentence now reads: "Fluorescent far field optical microscopy detects light emitted by fluorophores in a specimen using detectors located at certain distance from the sample. This approach is currently the basis of the most biological applications of SRM [8-10]."
As you suggested in line 402 to discuss the phospho-CTD specific antibodies, we have added: "It is worth mentioning that the antibody used in Fig. 2 recognizes all RPB1 CTD forms, however, a plethora of antibodies characteristic for specific stages of transcription exists. Human RPB1 CTD contains 52 heptad repeats that can be reversibly phosphorylated and the specific phosphorylation pattern reflects subsequent steps of transcription [123-125]. Importantly, distinct steps of transcription could be therapeutically targeted in cancer and other complex diseases [126]. Immunolabeling with antibodies against distinct RPB1 CTD phosphorylations results in distinct spatio-temporal localization of RPB1 and reflects distinct stages of transcription. Nevertheless, multiple antibodies can bind to one RPB1 CTD due to its repetitive organization." This section then contains several new references:
[123] Hsin JP, Manley JL. The RNA polymerase II CTD coordinates transcription and RNA processing. Genes Dev. 2012;26(19):2119-2137. doi:10.1101/gad.200303.112
[124] Heidemann M, Hintermair C, Voß K, Eick D. Dynamic phosphorylation patterns of RNA polymerase II CTD during transcription. Biochim Biophys Acta. 2013;1829(1):55-62. doi:10.1016/j.bbagrm.2012.08.013
[125] LeBlanc BM, Moreno RY, Escobar EE, Venkat Ramani MK, Brodbelt JS, Zhang Y. What's all the phos about? Insights into the phosphorylation state of the RNA polymerase II C-terminal domain via mass spectrometry. RSC Chemical Biology. 2021. doi: 10.1039/D1CB00083G.
We have also included a new important and relevant reference:
[126] Martin RD, Hébert TE, Tanny JC. Therapeutic Targeting of the General RNA Polymerase II Transcription Machinery. Int J Mol Sci. 2020;21(9):3354. Published 2020 May 9. doi:10.3390/ijms21093354
Moreover, to the last section we have added previously forgotten and relevant reference: "So far, we started to paint with nanometer precision the static pictures of nuclear PIP distributions [119, 127]."
[127] Hoboth, P, Šebesta, O, Sztacho M, Castano E, Hozak, P. : Dual-color dSTORM imaging and ThunderSTORM image reconstruction and analysis to study the spatial organization of the nuclear phosphatidylinositol phosphates. MethodsX. 2021(8)101372-. doi: 10.1016/j.mex.2021.101372.
Finally, we regret that in the original version of our manuscript we forgot to include the acknowledgement, which we have added in the revised version: "Construct H2B-Halo-SNAP and Janelia Fluor (JF)549 SNAP tag substrate used in Figure 1 were kind gifts of Xavier Darzacq from UC Berkeley and Luke Lavis from Janelia Research Campus of the Howard Hughes Medical Institute, resp. We would like to thank to Ivan Novotný and Helena Chmelová ..."
Please, find attached the revised version of our manuscript with marked changes and we're looking forward to hearing back from you.
With kind regards,
Authors.

Reviewer 2 Report
This paper reviews the field of superresolution from the very beginning. The authors describe a number of techniques and illustrate them by examples related to transcription. The paper is generously illustrated and very clearly written. Just two typos:
l. 96 switch of - switch off
l.334 heterichromatin - heterochromatin
Author Response
Dear Reviewer,
We appreciate your positive evaluation and thank you for pointing out the typos, which we have corrected in the revised version of our manuscript.
Thank you,
Kind regards,
Authors.
